# Molecular Effect of Variants in Toll-like Receptor 4 Gene in Saudi Patients with Type 2 Diabetes Mellitus

**DOI:** 10.3390/cells12192340

**Published:** 2023-09-23

**Authors:** Zeina S. Alkudmani, Amal F. Alshammary, Imran Ali Khan

**Affiliations:** Department of Clinical Laboratory Sciences, College of Applied Medical Sciences, King Saud University, Riyadh 11433, Saudi Arabia; zalkudmani@ksu.edu.sa (Z.S.A.); aalshammary@ksu.edu.sa (A.F.A.)

**Keywords:** T2DM, *TLR4* gene, rs11536889, rs4986790, rs4986791, Saudi population

## Abstract

Single-nucleotide polymorphisms (SNPs) in the Toll-like receptor 4 (*TLR4*) gene have been documented in type 2 diabetes mellitus (T2DM) and other diseases in the Saudi population. We investigated the relationship between rs11536889, rs4986790, and rs4986791 SNPs in the *TLR4* gene and T2DM in the Saudi population; 105 patients with T2DM and 105 healthy controls were analyzed. The *TLR4* gene was amplified through PCR, followed by restriction fragment length polymorphism analysis for rs4986791 and Sanger sequencing for rs11536889 and rs4986790 SNPs. The clinical and biochemical characteristics were associated with T2DM (*p* < 0.05). The rs11536889, rs4986790, and rs4986791 SNPs in control subjects followed the Hardy–Weinberg equilibrium (*p* > 0.05). Alleles were associated with rs11536889, rs4986791, heterozygous codominant, and dominant models (*p* < 0.05). However, the rs4986790 SNP was not associated with T2DM (*p* > 0.05). Logistic regression analysis showed that high-density lipoprotein cholesterol (HDLc) levels were associated with T2DM (*p* < 0.001). Analysis of variance showed that waist (*p* = 0.0005) and hip circumferences (*p* = 0.002) in rs4986790 and rs4986791 SNPs, in SBP (*p* = 0.001), DBP (*p* = 0.002), and HDLc levels (*p* = 0.003), were associated with T2DM subjects. T2DM was also associated with the haplotype (*p* < 0.001) but not with linkage disequilibrium. The gene–gene interaction was associated with the three SNPs studied in patients with T2DM according to the generalized multifactor dimensionality reduction model (*p* < 0.0001). Dendrogram and graphical depletion analysis revealed a moderate association in patients with T2DM. The results suggest that rs11536889 and rs4986790 SNPs are genotypically and allelically associated with T2DM in Saudi patients. Future functional studies are recommended to validate the genetic roles of these SNPs in the pathogenesis and progression of diseases.

## 1. Introduction

Diabetes mellitus (DM) is a clinical syndrome characterized by hyperglycemia at different states, such as fasting, postprandial, and oral glucose tolerance tests [1]. The impact of blood glucose levels on human health is often underestimated. Diabetes develops when the connection between the pancreas and insulin is disrupted and when insulin is not used [2,3]. The first mention of DM can be found in ancient Egypt. The Ebers papyrus, which dates to 1550 BC, was found in a cemetery in the southern Egyptian city of Thebes in 1862 [4]. Furthermore, DM presents as insulin-dependent/type 1 diabetes mellitus (T1DM), which is characterized by the autoimmune destruction of pancreatic islet β-cells, and non-insulin-dependent/type 2 diabetes mellitus (T2DM), which is caused by inadequate insulin production and insulin resistance (IR) and is an endocrine and metabolic disease [5,6]. Cardiovascular disorders (CVD) are the common cause of 70% of death in patients with T2DM, accompanied by a combination of hyperglycemia, dyslipidemia, and IR [7]. The World Health Organization (WHO) has estimated the prevalence of T2DM in the Saudi population among adults aged 18 at 14.4%, which increased to 18.3% in 2019 among Saudi adults aged 20–79. Previous studies have suggested a high prevalence of CVD risk factors in Saudi Arabia [8]. Obesity and age are considered risk factors for the increasing incidence of T2DM in the Gulf Cooperation Council region. The prevalence of obesity among people aged 25 years and older was expected to reach 41 and 78% in men and women, respectively, by 2022 [9], whereas the incidence of T2DM increased from 8.5 to 39.5% between 1992 and 2022.

Infectious diseases are the second leading cause of mortality worldwide. T2DM increases the susceptibility to infections initiated by hypoglycemia and ketoacidosis [10,11]. The immune system is a mediator of the connection between DM and infections. DM can weaken the immune system, reducing its capacity against infections, and increased glucose levels can impair the function of white blood cells. Infections have also been documented as an adverse effect in individuals with DM and high IR levels [11,12]. Single-nucleotide polymorphisms (SNP) can be found in the coding and noncoding sections of the genome and may determine the susceptibility, severity, or protection from diseases. The catalytic function, stability, and expression level of a protein can be affected via SNPs [13]. Molecular biology research has provided evidence of the association of genetic variants or SNPs with T2DM through genome-wide association studies [14]. The molecular pattern recognition performed by the innate immune system is facilitated by Toll-like receptors (TLR) [15]. Among the ten subtypes of TLRs, Toll-like receptor 4 (TLR4) is the first line of defense and regulates the immune system by identifying pathogen-associated molecular patterns and activating immune response genes. Genetic changes in *TLR4* have been linked to an increased risk of developing T2DM [16]. Furthermore, earlier research identified *TLR4* as a potentially significant factor in regulating susceptibility to T2DM [17]. The chromosomal location of *TLR4* is 9q33.1 and it has three exons and two introns. TLR4 is expressed in a limited number of cells, including the pancreas and adipose tissues [18]. Previous studies have identified an inflammatory response and upregulation of TLR4 in IR, obesity, and T2DM. Moreover, SNPs of the *TLR4* gene can participate in the development of T2DM disease [16]. Globally, results regarding the relationship between *TLR4* SNPs and human diseases have been inconsistent, and few studies have been performed in Saudi Arabia and none on T2DM Saudi patients. Therefore, this study was designed to analyze single amino acid substitutions SNPs in the *TLR4* gene, aiming to analyze the molecular connection between rs11536889, rs4986790, and rs4986791 SNPs and T2DM risk in Saudi individuals.

## 2. Materials and Methods

### 2.1. Importance of Patients with T2DM

This case–control study was designed in the Department of Clinical Laboratory Sciences at King Saud University (KSU). We prospectively enrolled patients with T2DM who visited the outpatient clinic of KSU Hospital on regular and follow-up bases. The study population was selected from Riyadh, the capital of Saudi Arabia. Patients with T2DM who visited the hospital’s outpatient clinic were enrolled. The presence of T2DM was initially confirmed and validated using medical records, medical prescriptions, and laboratory records. Glucose levels were measured based on the recommendations of the American Diabetes Association criteria (ADA), which are defined as follows: diagnosis of diabetes is defined as fasting blood glucose (FBG) levels exceeding ≥7.0 mmol/L or 2 h postprandial blood glucose (PPBG) levels of ≥11.1 mmol/L [19]. The glycated hemoglobin (HbA1c) level was ≥6.5% in patients with T2DM. The inclusion criteria for patients with T2DM were based on the ADA criteria and patients whose glucose values were not elevated with the normal values; then, the patients were excluded from the T2DM group criteria and confirmed as controls after verifying their medical records for not having had any abnormal values documented in their glucose or HbA1c levels. The exclusion criteria for controls were based on elevated glucose levels and a diagnosis of any other chronic diseases. All patients recruited in this study were confirmed as Saudi natives based on their national identification documents. Based on the sample size formula, 210 Saudi subjects were selected, of which 105 were confirmed to have T2DM, and the remaining 105 were healthy controls.

### 2.2. Sample Size

The sample size of this study was reconfirmed from previous studies [20,21] where 203 samples were used, and 210 samples were recruited in this study. The sample size was determined following the equation:(1)$$n=\left(Z_{\alpha /2}+Z_{\beta }\right)*2^{2}*2^{\left(\sigma *2)/(d*2\right)}$$

### 2.3. Ethical Issues

We clarified all ethical issues, and one of the project parts of this manuscript received ethical approval from the Institutional Review Board at the Medical College of KSU. The concept of this study was explained to the participants, who attended the outpatient clinic on a regular follow-up basis. After signing the patient consent form, 210 patients were recruited, and the study was designed according to the Declaration of Helsinki.

### 2.4. Subject Details

We recorded limited details of the participants in the questionnaire, such as age, sex, body mass index (BMI), waist circumference, hip circumference, and hypertension (HTN) values. Higher HTN values were recorded as systolic blood pressure (SBP) and lower values as diastolic blood pressure (DBP). In addition, a family history of T2DM was also recorded, which is presented in the clinical details of T2DM cases and healthy controls.

### 2.5. Sample Collection and Laboratory Investigation

A total of 1050 mL of peripheral blood was collected from 210 participants in 210 coagulant and 210 anticoagulant tubes. Each patient had 5 mL of blood drawn; 2mL for molecular/HbA1c analysis and 3 mL for serum analysis. Serum samples were collected for FBG, PPBG, and lipid profile parameters, which consisted of triglycerides (TG), different cholesterols, total cholesterol (TC), and high- and low-density lipoprotein cholesterol (HDLc and LDLc, respectively). Biochemical analysis was performed using COBAS automated equipment with ROCHE kits. However, the LDLc levels were calculated. In addition, HbA1c levels were measured in EDTA blood.

### 2.6. Molecular Investigation

EDTA blood was used to extract genomic DNA from 210 samples using a Qiagen DNA isolation kit (Hilden, Germany), following the manufacturer’s protocol. The quantity of DNA in the 210 samples was measured using a NanoDrop spectrophotometer (Thermo Fischer Scientific, Waltham, MA, USA). Different concentrations of DNA were measured, and all the DNA samples were concentrated to 20 µg/mL and stored at −80 °C for polymerase chain reaction (PCR) analysis. In this study, SNPs rs11536889, rs4986790, and rs4986791 were selected from the *TLR4* gene, and genotyping was performed using PCR analysis. The process of PCR started with a 50 µL reaction in which 10× buffer, Taq DNA Polymerase, MgCl_2_, dNTPs, primers (forward and reverse), and DNA templates were included, and double-distilled water was added to achieve the final volume. The PCR master mix was prepared for 100 samples and a 96-well plate was used to prepare 96 samples. PCR was performed with a thermal cycler using 95 °C for 5 min as the initial denaturation step, 95 °C for 30 s for denaturation, 30 s for annealing, 72 °C for 45 s for extension, and 72 °C for 5 min for the final extension. The different annealing temperatures were recorded for each SNP as follows: 64 °C for rs11536889, 62 °C for rs4986790, and 60 °C for rs4986791. PCR was performed for 35 cycles at 4 °C. The time required to complete all 3 SNPs was 1.24–1.32 h. The PCR products were subjected to 2% agarose gel electrophoresis and stained with ethidium bromide to confirm band sizes of 319, 200, and 106 bp for rs11536889, rs4986790, and rs4986791 SNPs, respectively. The rs4986791 SNP was digested with *Hinf*I (GAA^↑^CC-GAA^↑^TC) enzyme for 2 h at 37 °C. The band sizes of rs4986791 (+1196 C/T) were 106 and 87/19 bp for the CC and TT genotypes, respectively. The heterogeneous (CT) genotype was 106/87/19 bp (Appendix A; Appendix A). The digested products were electrophoresed on a 3% agarose gel stained with ethidium bromide, and images were captured with UVI gel documentation. Molecular analyses were performed at the G-141 laboratory of the CLS Department. Sanger sequencing analysis was performed outside the G-141 laboratory for the rs11536889 and rs4986790 SNPs. The Sanger dideoxy chain termination method was performed using the PCR products, a big terminal dye, 10 pM of forward and reverse primers, and a genetic analyzer. Multicapillary sequencing was performed using a bidirectional process. ABI, FASTA, and PDF files were obtained after sequencing, and chromatograms were analyzed using primer sequences for rs11536889 and rs4986790 SNPs (Appendix A).

### 2.7. Statistical Analysis

Categorical variables are expressed as total numbers and percentages. The mean and standard deviation (M ± SD) were applied to quantitative variables, and total numbers and percentages were used to express the qualitative variables. The differences between patients with T2DM and controls were tested using independent sample t-tests and the Mann–Whitney U test. The Hardy–Weinberg equilibrium (HWE) test was performed to confirm that the control group was representative of the population, as measured using goodness-of-fit χ^2^. Using logistic regression analysis with SNPstats software, odds ratios (ORs), *p*-values, and 95% confidence intervals (95%CIs) were calculated to analyze the risk and association of alleles, genotypes, and various genetic models with patients with T2DM and control individuals. Yates correction was used for the absence of genotypes present in all 3 SNPs of the *TLR4* gene in both the controls and T2DM cases. Logistic regression analysis was performed between the dependent and independent variables using SPSS software for patients with T2DM. The association between the 3 SNPs and T2DM covariates was studied via one-way ANOVA using Jamovi software [22]. Haploview software (version 4.2) was used to conduct the linkage disequilibrium haplotype and simultaneous functional prediction analyses [23]. The generalized multifactor dimensionality reduction (GMDR) model [24] was used to investigate gene–gene interactions, dendrograms, and graphical depletion models using the 3 SNPs of the *TLR4* gene. Statistical significance was defined as *p* ≤ 0.05.

## 3. Results

### 3.1. Demographic Profile for Performing Student t-Tests between T2DM and Control Subjects

Based on the inclusion and exclusion criteria, 105 patients with T2DM and 105 controls were recruited. Table 1 presents the stratified, clinical, and baseline information for both groups. Patients with T2DM were older and statistically associated compared to controls (63.68 ± 8.43 vs. 52.85 ± 7.97 years; *p* < 0.0001). Patients with T2DM comprised 47.62 and 52.38% of males and females, respectively, whereas 62.86 and 37.14% of the control subjects were males and females, respectively. The mean weight (79.65 ± 13.66 vs. 76.48 ± 11.97 Kg; *p* = 0.07), BMI (31.16 ± 5.13 vs. 29.44 ± 4.39; *p* = 0.009), SBP (126.50 ± 10.66 vs. 116.60 ± 6.26; *p* < 0.0001), DBP (79.82 ± 6.27 vs. 77.22 ± 5.43; *p* = 0.001), FBG (12.91 ± 4.32 vs. 5.44 ± 0.47; *p* < 0.0001), Hb1Ac (7.22 ± 0.89 vs. 5.36 ± 0.41; *p* < 0.0001), TG (2.09 ± 5.75 vs. 1.80 ± 0.99; *p* = 0.61), TC (5.75 ± 1.18 vs. 5.17 ± 0.92; *p* = 0.01), HDLc (1.39 ± 1.17 vs. 0.67 ± 0.30; *p* < 0.0001), and LDLc levels (3.81 ± 0.97 vs. 3.68 ± 0.80 ; *p* < 0.0001) were significantly higher in T2DM cases than in the controls. Height (159.72 ± 8.42 vs. 161.23 ± 7.13; *p* = 0.16), waist circumference (96.59 ± 16.77 vs. 98.81 ± 18.83; *p* = 0.36), and hip circumference (103.88 ± 15.57 vs. 105.90 ± 16.79; *p* = 0.38) were found to have values in controls in comparison with T2DM cases. However, the *p*-values were not significantly different between patients with T2DM and controls (*p* > 0.05). Finally, all patients with T2DM had a family history of T2DM, while only 43.8% of the controls had a family history of T2DM (*p* < 0.0001).

### 3.2. Verification of Enrolled Population Analysis

The genotype distributions of rs11536889 (*p* = 0.84), rs4986790 (*p* = 0.96), and rs4986791 SNPs (*p* = 0.11) in the control group were all HWE (*p* > 0.05). However, in T2DM cases, the rs11536889 (*p* = 0.68) and rs4986790 (*p* = 0.80) SNPs were consistent with HWE, while rs4986791 (*p* = 0.003) was not in agreement in HWE in T2DM cases. Additionally, variant allele frequencies and chi-square (χ^2^) values were shown in Table 2. Finally, based on the HWE association in control subjects, this study can be conducted to perform a molecular analysis of T2DM cases and control subjects.

### 3.3. Genetic Association of TLR4 Gene SNPs with T2DM and Controls 

Table 3 presents the genotype distribution of the studied SNPs in the *TLR4* gene between T2DM cases and control subjects using ORs, 95%CIs, and *p* values in codominant models of homozygous and heterozygous as well as in genetic models such as dominant, codominant, and recessive models for each SNP. Among the rs11536889 SNP, the GG, GC, and CC genotypes observed in T2DM and control individuals were 84.7%, 14.3%, 0.9%, 96.2%, and 3.8%, respectively. The GC genotype (GC vs. GG: OR—4.25 (95%CI: 1.36–13.29); *p* = 0.007) and dominant model (GC + CC vs. GG: OR—4.53 (95%CI: 1.46–14.08); *p* = 0.004) were substantially associated with a fourfold increase in T2DM risk. The homozygous codominant genotype (CC + CC vs. GG: OR—3.41 (95%CI: 0.13–84.56); *p* = 0.42) and genetic models (GG + CC vs. GC: OR—0.23 (95%CI: 0.07–0.74); *p* = 0.008 and GC + CC vs. GG: OR—0.49 (95%CI: 0.01–14.92); *p* = 0.67) could not show the genetic association after performing the Yates correction. The rs4986790 SNP in *TLR4* gene was not associated with any of the genotypes (AG vs. AA: OR—5.21 (95%CI: 0.59–45.29); *p* = 0.09 and GG vs. AA: OR—1.00 (95%CI: 0.01–50.85); *p* = 0.99) or genetic (AG + GG vs. AA: OR—1.00 (95%CI: 0.01–50.85); *p* = 0.99; AA + GG vs. AG: OR—0.19 (95%CI: 0.02–1.67); *p* = 0.09 and AA+ AG vs. GG: OR—1.00 (95%CI: 0.01–50.85); *p* = 0.99) models after applying Yates correction for the absence of genotypes in studied groups. The final SNP rs4986791 in the *TLR4* gene was associated with a combination of codominant heterozygous (CT vs. CC: OR—3.34 (95%CI: 1.87–5.99); *p* = 0.0003) and dominant models (CT + TT vs. CC: OR—3.52 (95%CI: 1.97–6.29); *p* = 0.0001), which dramatically increased the risk of T2DM by approximately 3.4 times. Other genotype (TT vs. CC: OR—11.67 (95%CI: 0.58–230.0); *p* = 0.04) and genetic models (CC + TT vs. CT: OR—0.31 (95%CI: 0.17–0.56); *p* = 0.0008 and CC + CT vs. TT: OR—0.31 (95%CI: 0.17–0.56); *p* = 0.13) failed to show the association after adding Yates correction for the absence of genotype.

### 3.4. Allelic Association Studies

We observed significant differences in the allelic distributions of rs11536889 and rs4986791 SNPs between patients with T2DM and controls. The C allele in the rs11536889 SNP was found in 8.1 and 1.9% of T2DM cases and controls, respectively, and the G allele in 98.1 and 91.9% of controls and patients with T2DM, respectively. The T allele in the rs4986791 SNP was present in 29.5 and 13.3% of T2DM cases and controls, respectively, and 86.7 and 70.5% of the C alleles were present in patients with T2DM and controls, respectively. The logistic regression analysis confirmed the association of the C allele of rs11536889 (C vs. G: OR—4.53 (95%CI: 1.51–13.72); *p* = 0.003) and the T allele of rs4986791 SNPs (T vs. C: OR—2.72 (95%CI: 1.65–4.47); *p* = 0.00005) with a high risk of T2DM. The G allele frequencies in T2DM cases and controls were 2.4 and 0.5%, respectively, while A allele frequencies in T2DM cases and controls were 97.6 and 99.5%. There was no association between G and A alleles in T2DM cases (G vs. A: OR—5.09 (95%CI: 0.59–44.1); *p* = 0.100). The allele results are presented in Table 4.

### 3.5. Association between TLR4 SNPs and T2DM Covariates 

After completing the genotype and allele frequency distributions, we studied the association between *TLR4* SNPs and T2DM covariates using a logistic regression model (LRM), as shown in Table 5. In this study, 14 T2DM covariates were studied using rs11536889, rs4986790, and rs4986791 SNPs, and the HDLc covariate was strongly associated (*p* < 0.0001). This shows that HDLc levels are associated with LRM.

### 3.6. Analysis of ANOVA Examined between Dependent and Independent Variables

In this study, 14 covariates were identified as dependent variables, including age, weight, BMI, waist circumference, hip circumference, SBP, DBP, FBG, HbA1c, TG, TC, HDLc, and LDLc levels, and rs11536889, rs4986790, and rs4986791 SNPs were considered independent variables, as shown in Table 6. Among the three SNPs, high levels were found in different covariates or dependent variables such as age (75.00 ± 1.00), weight (83.87 ± 21.01), BMI (32.36 ± 7.04), waist circumference (122.33 ± 45.71), hip circumference (120.00 ± 1.00), SBP (150.00 ± 1.00), DBP (82.60 ± 5.17), FBG (13.58 ± 4.64), HbA1c (7.82 ± 0.40), TG (3.64 ± 1.69), TC (5.85 ± 1.16), HDLc (1.88 ± 2.00), and LDLc levels (4.09 ± 0.94) in different independent variables. However, there was no statistically significant association between the dependent or independent variables in the rs11536889 SNP (*p* > 0.05). In the rs4986790 SNP, waist circumference (*p* = 0.0005) was strongly associated, whereas with the rs4986791 SNP, hip circumference (*p* = 0.002), SBP (*p* = 0.001), DBP (*p* = 0.006), and HDLc levels (*p* = 0.003) were associated.

### 3.7. Haplotype Interference Analysis 

Haplotype interference analysis was performed between the combination of T2DM cases and controls as well as rs11536889, rs4986790, and rs4986791 SNPs. Haplotype interaction details are listed in Table 7. The combination of T2DM and control subjects in the Saudi population among the three SNPs was tested against the combination and probable predictors. The outcomes of the five measures and the combination of G–A–T (OR—4.27 (95%CI: 2.02–9.01); *p* < 0.0001) and A–A–C (OR—8.46 (95%CI: 1.40–51.12); *p* = 0.02) showed the strongest association.

### 3.8. Association of Linkage Disequilibrium 

Table 8 describes the combination of patients with T2DM and control subjects who were subjected to LD analysis for rs11536889, rs4986790, and rs4986791 SNPs. The overall analysis revealed that LD did not play a role in the combination of the rs11536889, rs4986790, and rs4986791 SNPs in *TLR4*, as shown in Figure 1.

### 3.9. Association of Gene–Gene Interaction Analysis Studied via GMDR Model

The samples obtained from patients with T2DM and control subjects in the present study were not significantly different in terms of gender matching (*p* = 0.78). Table 9 describes the combination of gene–gene interactions between the rs11536889, rs4986790, and rs4986791 SNPs in patients with T2DM. Model 1 represents rs11536889, model 2 represents the combination of rs11536889 and rs4986791, and model 3 represents the combination of rs11536889, rs4986790, and rs4986791. Individual associations were found in all three models with a 10/10 CVC ratio. The OR for rs11536889 SNP was OR—3.52 (95%CI: 1.97–6.29; *p* < 0.0001), for rs4986790 it was OR—4.79 (95%CI: 2.67–8.59; *p* < 0.0001), and for rs4986791 it was OR—4.98 (95%CI: 2.77–8.93; *p* < 0.0001). Gene–gene interactions were strongly associated with the three SNPs in T2DM cases. Dendrogram interaction graphs generated using hierarchical cluster analysis show the presence, strength, and nature of epistatic effects. The greater the interaction, the more the line linking the two components moves to the right. In general, red color indicates a high degree of synergy when present between two lines. In this study, among SNPs, brown and blue color lines were present, indicating nominal and redundant interactions. Figure 2 shows the dendrogram, and graphical depletion is shown in Figure 3. In general, darker cells indicate a riskier combination, and light cells indicate a low-risk or no risk combination. The absence of genotype data was indicated by white/blank cells. The overall analysis of this study indicated the risk associated with the combination of CC genotypes in rs11536889 and AG genotypes in rs4986790 SNPs, indicating that the darker cells and rest of the cells were lighter or white/blank, indicating that there is no effect.

## 4. Discussion

Previously, multiple genome-wide association studies (GWAS), candidate genes, and linkage analysis studies investigated the relationship between genetic SNPs and the risk of T2DM, which were then utilized to identify high-risk individuals. GWAS began being used for screening patients with T2DM in 2000, and by 2016, over 60 SNPs of Asian and European ancestry had been identified [25,26]. TLR started being used in 1998 under the Sapporo criteria, with which antiphospholipid syndrome (APS) was proposed. Demographic studies have discovered a limit of 40–50 cases per 100,000 individuals in the general population. APL begins to interact with several cell types, ranging from endothelial cells to trophoblast cells, resulting in the activation of the coagulation and complement systems as well as the suppression of fibrinolytic processes by aPLs via cell surface receptors, including TLR, which are members of the pattern recognition receptor family and play a crucial role in the innate immune system [27]. TLRs are classified into ten types, with TLR2 and TLR4 responsible for detecting Gram-positive and Gram-negative bacteria, respectively [28]. TLR4 and T2DM are linked to abnormal innate and adaptive immunity, which is present in β-cells in individuals with T2DM, elevated BMI, or both. TLR receptors were discovered in pancreatic islet cells, and TLR4 is upregulated in response to high glucose levels as well as obesity [29]. Multiple molecular studies have been performed on SNPs present in the *TLR4* gene, and limited studies have documented different forms of diabetes. In Saudi Arabia, molecular screening for the hepatitis C virus (HCV), glaucoma, leukemia, cancer, and CVD has been undertaken [30,31,32,33,34,35,36,37,38,39]. However, no studies on T2DM have been conducted in the Saudi population, although there is an association between infection, the *TLR4* gene, and T2DM. Therefore, the purpose of this study was to examine the molecular SNPs rs11536889, rs4986790, and rs4986791 in the *TLR4* gene in patients with T2DM within the Saudi population. The results of the current study indicate that genotypes, genetic models, and allele frequencies were significantly associated with rs11536889 and rs4986791 SNPs in patients with T2DM (*p* < 0.05). There was no association between any of the forms of the rs4986790 SNP (*p* > 0.05). The HWE analysis was in agreement with all the studied SNPs in the control subjects (*p* > 0.05). Further statistical analysis revealed positive associations between the dependent and independent factors. The purpose of performing LRM was to predict the relationship between the dependent and independent variables, and in this study, HDLc levels (*p* < 0.0001) were strongly associated. ANOVA analysis showed a strong association with waist circumference (*p* = 0.0005) in the rs4986790 SNP, whereas for rs4986791 SNP, hip circumference (*p* = 0.002), SBP (*p* = 0.001), DBP (*p* = 0.006), and HDLc levels (*p* = 0.003) were associated. Haplotype analysis showed strong association with G–A–T (OR—4.27 (95%CI: 2.02–9.01); *p* < 0.0001) and A–A–C (OR—8.46 (95%CI: 1.40–51.12); *p* = 0.02), whereas LD analysis revealed a negative association. Among the GMDR models, gene–gene interactions showed strong association (*p* < 0.0001) with a 10/10 CVC. The dendrogram indicates nominal and redundant interactions, and graphical depletion indicates the risk associated with the combination of CC genotypes in rs11536889 and AG genotypes in rs4986790 SNPs.

Ten studies have documented the rs4986790 and rs4986791 SNPs’ associations with different human diseases in Saudi Arabia [30,31,32,33,34,35,36,37,38,39]. In addition to the rs4986790 and rs4986791 SNPs, other SNPs, such as rs1927906, rs7856729, rs2770150, rs10759931, and rs10759932, were studied. The rs4986790 SNP was not associated with any of the studied populations in Saudi Arabia [30,31,32,33,34,35,36,37,38,39], and the results of this study agreed with previous studies in Saudi Arabia. The rs4986791 SNP was associated with pulmonary tuberculosis (PTB; *p* < 0.001) [32] and *Helicobacter pylori*-related gastric disease (*p* = 0.003) [37]. Both rs4986790 and rs4986791 SNPs were shown to have a significant molecular role in PTB in the Saudi population. The AG and GG genotypes were found in 18.3 and 15% of control cases, and 13.3 and 18.3% of patients with PTB (*p* > 0.05), respectively. The CT and TT genotypes were found in 26.7 and 65.9% of control cases, and 21.7 and 48.3% in patients with PTB, respectively (*p* > 0.05). No molecular studies have documented the rs11536889 SNP in the Saudi population, and our study was the first to explore the fourfold increase in genotype risk associated with rs11536889 SNP in patients with T2DM (*p* < 0.05).

In this study, the frequency of codominant homozygous genotypes was extremely low in patients with T2DM, with no codominant homozygous genotypes present in control subjects. In the rs11536889 SNP, only a single CC genotype was present in patients with T2DM, and in the rs4986791 SNP, 2.9% of the TT genotypes were present in T2DM cases. However, none of the GG genotypes were present in the T2DM or control cases. In the rs4986791 SNP, a high frequency of TT genotypes was present in the cohort studied by Fouad et al., with 48.3% in PTB and 26.7% in control subjects. Furthermore, 1.2% of TT genotypes were present in patients with glaucoma in Mousa et al.’s studies, and 3.8 and 11.9% of TT genotypes were present in controls and patients with *Helicobacter pylori*-related gastric disease in Eed et al.’s studies, respectively; however, in Eed et al.’s studies, all the controls were Saudi natives. Our study controls had a mean age above 52 years, with overweight (29.44 ± 4.29), and most importantly, a family history of DM was found in approximately 44%. Here, we recorded only family histories of T2DM and HTN, and excluded other family histories, which is one of the limitations of this study. All 210 participants signed a patient consent form and belonged to Saudi nativity, which could be easily predicted or analyzed. Among the control subjects, those with the codominant homozygous genotypes in rs11536889, rs4986790, and rs4986791 SNPs were more susceptible to the disease, since the control population did not have any of the CC/GG/TT genotypes.

The rs11536889 SNP was studied in our population and showed a positive association; similar results were obtained in other ethnic populations in patients with T2DM [40] and limited studies have shown a negative association [41,42,43]. Both positive [44,45,46] and negative associations [16,47,48,49,50,51,52,53,54,55,56,57,58,59,60] have been reported for the rs4986790 SNP. Only a few studies [44,45] on the rs4986791 SNP are associated with T2DM disease, and other ethnic populations studied with the rs4986791 SNP showed a negative association [16,46,50,51,53,54,55,56,58,59,60]. However, the rs4986790 SNP has been studied in German, American, Finnish, Korean, Polish, Brazilian, Dutch, Croatian, Chinese, Indian, Bulgarian, Romanian, Turkish, and Iranian populations with T2DM [16,44,45,46,47,48,49,50,51,52,53,54,55,56,57,58,59,60]. For the rs11536889 SNP, Wang et al. [35] discovered 7.4 and 3.9% of CC genotypes, Huang et al. [36] found 6.3 and 5.1% of CC genotypes, and Peng et al. [42] found 4.9 and 6.3% of CC genotypes in patients with T2DM and controls, respectively. Li et al. [43] documented 6.3 and 5.3% TT genotypes in T2DM and T2DM with tuberculosis, respectively.

Regarding the rs4986790 SNP, Kolek et al. [48], Hernesniemi et al. [49], and Beijk et al. [52] performed a combined genotype analysis using a combination of the dominant model, that is, AG and GG genotypes. In addition to these three studies [48,49,52], Buraczynska et al. [46] reported a 0.46% prevalence of the GG genotype in T2DM cases and 0.13% in controls. According to Singh et al., 0.8% [39] and 0.52% [50] of patients with T2DM had the GG genotypes. In Indian and Romanian populations, the prevalence of GG genotypes in T2DM/controls was reported to be 3.01%/3.94% [52], 1.24%/0.76% [40], and 4.54%/5%, respectively [54]. However, some studies have found 0.87% [47] and 1% [51] of GG genotypes in the control population. In our study, the GG genotype was absent in T2DM cases and control subjects. A couple of studies have confirmed that 0.7% [46] and 0.3% [56] of the TT genotypes of the rs4986791 SNP were present only in patients with T2DM in the Polish population. The remaining studies corroborated the TT genotypes in patients with T2DM and controls at 4.8%/1.53% in Singh et al. [44], 0.8%/1.25% in Singh et al. [55], 0.6%/0.71% in Gond et al. [45], and 11.11%/6.5% in Aioanei et al. [59]. In our study, the TT genotype was present in 2.9% of patients with T2DM and was absent in control subjects. The overall conclusions of the rs4986790 and rs4986791 SNPs confirmed that the prevalence of the GG/TT genotypes is rare in limited ethnicities.

The first meta-analysis of rs4986790 and rs4986791 SNPs with T2DM was conducted by Assmann et al. [61] and confirmed that both SNPs exhibited a relationship with T2DM protection; in 2015, Yin et al. [62] studied an identical meta-analysis and found that both SNPs were not associated. Finally, Fan et al. [63], with updated case–control studies, performed a meta-analysis in T2DM subjects with rs11536889, rs4986790, and rs4986791 SNPs in the studied patients with T2DM. The individual analysis confirmed that rs11536889 SNP has a protective role in patients with T2DM in the Chinese population, rs4986790 SNP was associated with an increased risk of T2DM in Asian patients, and rs4986791 SNP was confirmed as having an increased risk in patients with T2DM in both Asian and Caucasian populations. A combined study of the SNPs rs11536889, rs4986790, and rs4986791 revealed that they may increase the risk of T2DM [63]. A recent meta-analysis evaluated the risk of diabetic microvascular complications and found that rs4986790 SNP was associated with T2DM, particularly in patients with diabetic retinopathy [64].

In our study, we identified a relationship between HDLc levels and the combination of three SNPs studied via a regression model (*p* < 0.001); ANOVA analysis validated the association between HDLc levels and the rs4986791 SNP in patients with T2DM (*p* = 0.003). The interleukin 1 receptor family includes TLRs. TLR4 was the first reported signaling receptor for lipopolysaccharides and is the most well-known member of this family. Heat shock proteins, fibronectin, fibrinogen, minimally modified and oxidized LDL, and free fatty acids are ligands with which TLR4 may interact, and they are all found at higher concentrations in T2DM subjects. TLR4 activation is believed to alter lipid metabolism, resulting in changes in HDLc levels. Furthermore, the inflammation caused by TLR4 activation may affect lipid metabolism and total cholesterol control. Previous studies have explored the potential impact of TLR4 and its SNPs on lipid metabolism and cholesterol levels, including HDLc [65,66]. Based on previous studies, we confirmed that SNPs in TLR4 affect HDLc and TC levels in different chronic diseases.

A family history of diabetes refers to the presence or history of diabetes in one or more close blood relatives, including parents, siblings, or children. This reveals a hereditary susceptibility to the disease and implies that other family members may be at a higher risk of developing diabetes. A family history of diabetes does not ensure that a person develops diabetes. Nonetheless, this indicates a larger risk of medical issues when compared to persons without a family history of diabetes. Both inherited variables and shared living practices within families increase risk. Individuals may be questioned about the existence of diabetes in their immediate family members while discussing their family history with healthcare experts or genetic counselors to estimate their risk and adopt the necessary preventative measures. Understanding family history is important because it may lead to earlier identification, better treatment, and lifestyle changes that lower the chances of acquiring diabetes or its consequences. Saudi Arabia has an increasing family history of diabetes, mainly due to lifestyle factors such as lack of exercise, poor diet, obesity, and overweight. Saudi Arabia has experienced significant socioeconomic development over the last four decades. Growth and affluence have significantly changed people’s lifestyles [67]. In this study, 43.8% of the controls and all patients with T2DM had a family history of DM.

The recruitment of a small sample size is one of the limitations of this study. Missing protein data are another limitation of this study. Finally, consanguinity and other family histories were not recorded. The strength of this study was the screening of the three SNPs in the *TLR4* gene.

## 5. Conclusions

In conclusion, this study confirmed that the rs11536889 and rs4986791 SNPs are associated with T2DM in the Saudi population, and additional statistical analysis confirmed the relationship between HDLc levels and the *TLR4* gene, which has been documented in previous studies. To the best of our knowledge, this work will contribute to understanding the role of the *TLR4* gene in patients with T2DM in the Saudi population. Future studies should investigate additional SNPs with a large sample size of cases, as well as the role that codominant homozygous genotypes have on the pathophysiology and development of T2DM.

## Figures and Tables

**Figure 1 cells-12-02340-f001:**
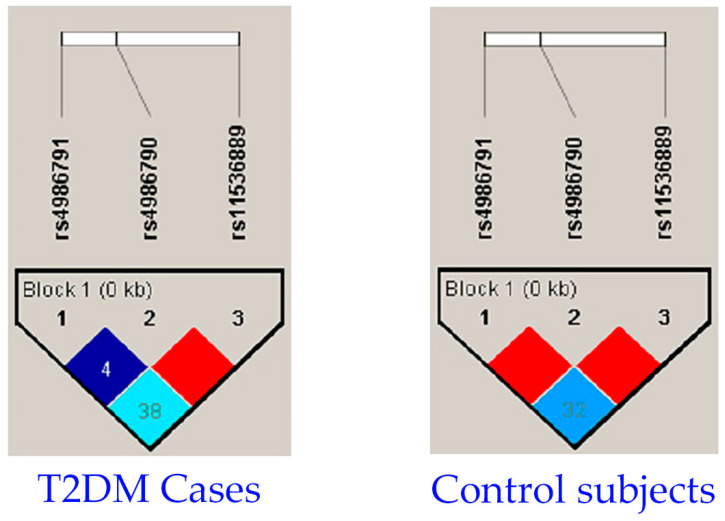
Analysis of linkage disequilibrium with rs11536889, rs4986790, and rs4986791 SNPs in *TLR4* gene. Legend: Haploview LD graph for rs4986791, rs4986790, and rs11536889 SNP studies in T2DM and control patients. Pairwise LD coefficient D was shown in both T2DM cases and control subjects. Standard color schemes of Haploview were applied for LD color display and red color indicates evidence of high LD, while other colors indicate evidence of moderate LD and white color indicates no evidence of LD. However, in this study, we observed red, different shades of blue, and turquoise. White was not observed in any LD cells.

**Figure 2 cells-12-02340-f002:**
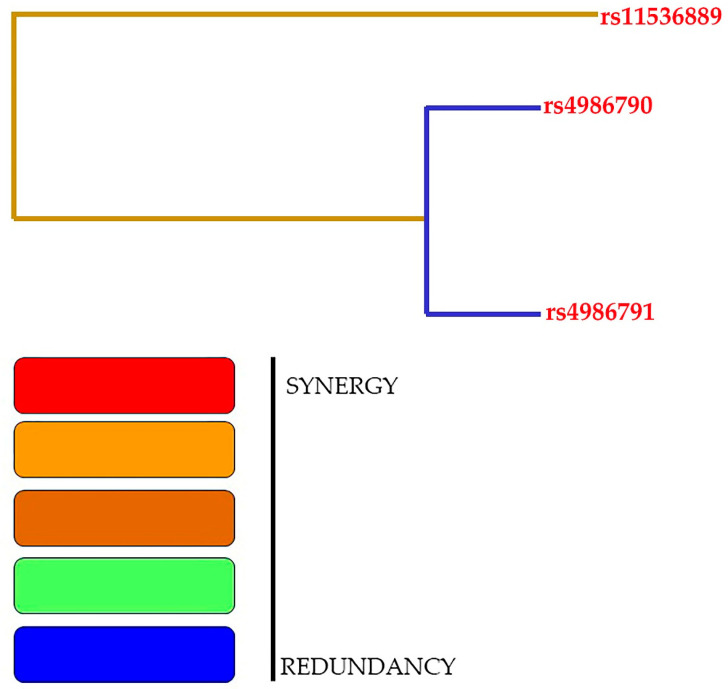
Representation of dendrogram using three SNPs involved in T2DM cases. Legend: the GMDR model analysis includes an interaction dendrogram.

**Figure 3 cells-12-02340-f003:**
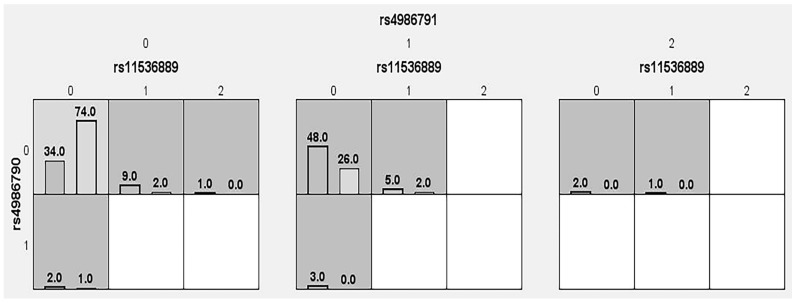
Graphical depiction of MDR analysis in TDM cases and SNPs in *TLR4* gene.

**Table 1 cells-12-02340-t001:** Clinical data obtained between T2DM and control subjects.

Covariates	T2DM Cases (n = 105)	Controls (n = 105)	*p* Value
Age (year)	63.68 ± 8.43	52.85 ± 7.97	<0.0001
Gender (male:female)	50 (47.62%):55 (52.38%)	66 (62.86%):39 (37.14%)	0.78
Weight (kilograms)	79.65 ± 13.66	76.48 ± 11.97	0.07
Height (centimeters)	159.72 ± 8.42	161.23 ± 7.13	0.16
BMI (kg/m^2^)	31.16 ± 5.13	29.44 ± 4.39	0.009
Waist	96.59 ± 16.77	98.81 ± 18.83	0.36
Hip	103.88 ± 15.57	105.90 ± 16.79	0.38
SBP (mmHg)	126.50 ± 10.66	116.60 ± 6.26	<0.0001
DBP (mmHg)	79.82 ± 6.27	77.22 ± 5.43	0.001
FBG (mmol/L)	12.91 ± 4.32	5.44 ± 0.47	<0.0001
Hb1Ac (%)	7.22 ± 0.89	5.36 ± 0.41	<0.0001
TG (mmol/L)	2.09 ± 5.75	1.80 ± 0.99	0.61
TC (mmol/L)	5.75 ± 1.18	5.17 ± 0.92	0.01
HDL-c (mmol/L)	1.39 ± 1.17	0.67 ± 0.30	<0.0001
LDL-c (mmol/L)	3.81 ± 0.97	3.68 ± 0.80	0.29
Family History of T2DM	105 (100%)	46 (43.8%)	<0.0001

**Table 2 cells-12-02340-t002:** HWE analysis observed in three SNPs of *TLR4* alleles in controls and T2DM cases.

SNPs	Minor Allele	Controls (n = 105)			T2DM (n = 105)		
		VAF	χ^2^	*p* Value	VAF	χ^2^	*p* Value
rs11536889	A	0.02	0.03	0.84	0.08	0.16	0.68
rs4986790	G	0.00	0.002	0.96	0.02	0.06	0.80
rs4986791	T	0.13	2.48	0.11	0.30	8.32	0.003

SNPs—single-nucleotide polymorphisms; VAF—variant allele frequencies; T2DM—type 2 diabetes mellitus.

**Table 3 cells-12-02340-t003:** Genotyping analysis for T2DM cases and controls.

Gene (rs Number)	Genotypes	T2DM Cases (n = 105)	Controls (n = 105)	OR (95%CI) and *p* Value
*TLR4* (rs11536889)	GG	89 (84.7%)	101 (96.2%)	Reference
	GC	15 (14.3%)	04 (3.8%)	OR—4.25 (95%CI: 1.36–13.29); *p* = 0.007
	CC	01 (0.9%)	00 (00%)	OR—3.41 (95%CI: 0.13–84.56); *p* = 0.42 *
	GC + CC Vs GG	16 (15.2%)	04 (3.8%)	OR—4.53 (95%CI: 1.46–14.08); *p* = 0.004
	GG + CC Vs GC	90 (85.7%)	101 (96.2%)	OR—0.23 (95%CI: 0.07–0.74); *p* = 0.008
	GG + GC Vs CC	104 (99.1%)	105 (100%)	OR—0.49 (95%CI: 0.01–14.92); *p* = 0.67 *
*TLR4* (rs4986790)	AA	100 (95.2%)	104 (99%)	Reference
	AG	05 (4.8%)	01 (1.0%)	OR—5.21 (95%CI: 0.59–45.29); *p* = 0.09
	GG	00 (00%)	00 (00%)	OR—1.00 (95%CI: 0.01–50.85); *p* = 0.99
	AG + GG Vs AA	05 (4.8%)	01 (1.0%)	OR—1.00 (95%CI: 0.01–50.85); *p* = 0.99
	AA + GG Vs AG	100 (95.2%)	104 (99%)	OR—0.19 (95%CI: 0.02–1.67); *p* = 0.09
	AA + AG Vs GG	105 (100%)	105 (100%)	OR—1.00 (95%CI: 0.01–50.85); *p* = 0.99
*TLR4* (rs4986791)	CC	46 (43.8%)	77 (73.3%)	Reference
	CT	56 (53.3%)	28 (26.7%)	OR—3.34 (1.87–5.99); *p* = 0.0003
	TT	03 (2.9%)	00 (0%) *	OR—11.67 (0.58–230.0); *p* = 0.04 *
	CT + TT Vs CC	59 (56.2%)	28 (26.7%)	OR—3.52 (1.97–6.29); *p* = 0.0001
	CC + TT Vs CT	49 (46.7%)	77 (73.3%)	OR—0.31 (0.17–0.56); *p* = 0.0008
	CC + CT Vs TT	102 (97.1%)	105 (100%) *	OR—0.31 (0.17–0.56); *p* = 0.13

* indicates Yates correction; OR—odds ratio; 95%CI—95% confidence interval.

**Table 4 cells-12-02340-t004:** Allele frequencies studied between T2DM and control subjects among *TLR4* gene.

Gene (rs Number)	Alleles	T2DM Cases (n = 105)	Control (n = 105)	OR (95%CI) and *p* Value
*TLR4* (rs11536889)	G	193 (91.9%)	206 (98.1%)	Reference
	C	17 (8.1%)	04 (1.9%)	OR—4.53 (1.51–13.72); *p* = 0.003
*TLR4* (rs4986790)	A	205 (97.6%)	209 (99.5%)	Reference
	G	05 (2.4%)	01 (0.5%)	OR—5.09 (0.59–44.1); *p* = 0.100
*TLR4* (rs4986791)	C	148 (70.5%)	182 (86.7%)	Reference
	T	62 (29.5%)	28 (13.3%)	OR—2.72 (1.65–4.47); *p* = 0.00005

OR—odds ratio; 95%CI—95% confidence interval.

**Table 5 cells-12-02340-t005:** Logistic regression studies in *TLR4* genetic variants and T2DM covariates.

Covariates	R-Value ^a^	Adjusted R Square Value	Standardized β-Coefficient for rs11536889	Standardized β-Coefficient for rs4986790	Standardized β-Coefficient for rs4986791	F	*p* Value ^b^
Age	0.067	−0.025	0.033	0.064	−0.004	0.154	0.927
Gender	0.066	−0.025	0.002	0.066	−0.007	0.149	0.930
Weight	0.130	−0.012	−0.039	0.119	0.023	0.581	0.629
BMI	0.069	−0.025	−0.002	0.053	−0.038	0.159	0.924
SBP	0.084	−0.022	−0.011	−0.004	0.084	0.241	0.868
DBP	0.110	−0.017	−0.029	0.082	−0.052	0.410	0.746
Waist	0.119	−0.015	−0.075	0.079	0.035	0.482	0.696
Hip	0.238	0.029	0.039	−0.058	0.219	2.029	0.115
FBG	0.253	0.036	−0.021	0.188	−0.146	2.298	0.082
Hb1Ac	0.112	−0.017	−0.026	−0.112	0.008	0.427	0.734
TC	0.217	0.019	−0.091	−0.139	0.146	1.657	0.181
TG	0.122	−0.014	−0.106	−0.078	−0.010	0.506	0.679
HDLc	0.717	0.500	0.701	−0.085	−0.046	35.714	<0.001
LDLc	0.149	−0.007	0.031	0.129	−0.068	1.409	0.162

^a^ Predictors (constants): *TLR4* gene (rs11536889, rs4986790, and rs4986791). ^b^ Dependent variables are listed in covariates.

**Table 6 cells-12-02340-t006:** ANOVA analysis was studied between *TLR4* gene SNPs with its covariances.

	*TLR4* (rs11536889)				*TLR4* (rs4986790)				*TLR4* (rs4986791)			
	GG (n = 89)	GC (n = 15)	CC (n = 01)	*p* Value	AA (n = 100)	AG (n = 05)	GG (n = 00)	*p* Value	CC (n = 46)	CT (n = 56)	TT (n = 03)	*p* Value
Age	63.57 ± 8.64	63.53 ± 7.04	75.00 ± 1.00	0.11	63.69 ± 8.42	63.40 ± 9.81	00.00 ± 00.00	0.94	64.22 ± 9.94	63.22 ± 7.22	62.00 ± 5.20	0.79
Weight	78.91 ± 12.11	83.87 ± 21.01	82.5 ± 1.00	0.42	79.63 ± 13.02	80.20 ± 25.62	00.00 ± 00.00	0.92	79.30 ± 14.88	80.62 ± 12.52	67.00 ± 12.17	0.23
BMI	30.96 ± 4.79	32.36 ± 7.04	31.43 ± 1.00	0.66	31.14 ± 5.05	31.68 ± 7.18	00.00 ± 00.00	0.81	30.59 ± 5.79	31.92 ± 4.40	25.66 ± 3.60	0.07
Waist	97.75 ± 14.45	88.89 ± 27.79	93.00 ± 1.00	0.17	95.42 ± 14.06	122.33 ± 45.71	00.00 ± 00.00	0.0005	98.53 ± 17.36	94.50 ± 16.73	94.33 ± 9.02	0.48
Hip	104.23 ±15.74	101.56 ± 15.07	120.00 ± 1.00	0.49	103.30 ± 15.67	116.67 ± 2.08	00.00 ± 00.00	0.06	99.64 ± 17.43	109.55 ± 11.94	98.00 ± 2.00	0.002
SBP	126.42 ± 10.70	125.15 ± 8.91	150.00 ± 1.00	0.07	126.38 ± 10.72	130.00 ± 10.00	00.00 ± 00.00	0.46	130.35 ± 10.34	123.18 ± 9.95	132.50 ± 10.61	0.001
DBP	79.27 ± 6.39	82.60 ± 5.17	80.00 ± 1.00	0.16	79.93 ± 6.29	76.67 ± 5.77	00.00 ± 00.00	0.25	82.54 ± 6.19	77.84 ± 5.80	80.00 ± 0.00	0.006
FBG	13.11 ± 4.43	11.83 ± 3.64	11.36 ± 1.00	0.53	12.90 ± 4.37	13.23 ± 3.32	00.00 ± 00.00	0.86	13.58 ± 4.64	12.47 ± 4.01	10.97 ± 4.45	0.31
Hb1Ac	7.27 ± 0.91	6.88 ± 0.77	7.20 ± 1.00	0.29	7.19 ± 0.90	7.82 ± 0.40	00.00 ± 00.00	0.12	7.39 ± 0.92	7.04 ± 0.82	7.77 ± 1.29	0.07
TG	2.15 ± 2.07	1.75 ± 1.00	2.25 ± 1.00	0.76	2.10 ± 1.98	2.01 ± 1.09	00.00 ± 00.00	0.92	1.71 ± 0.72	2.32 ± 2.51	3.64 ± 1.69	0.11
TC	5.85 ± 1.16	5.23 ± 1.18	5.13 ± 1.00	0.14	5.75 ± 1.20	5.80 ± 0.79	00.00 ± 00.00	0.93	5.77 ± 1.10	5.75 ± 1.26	5.60 ± 1.15	0.97
HDL-c	1.31 ± 0.97	1.88 ± 2.00	0.86 ± 1.00	0.19	1.41 ± 1.19	0.98 ± 0.91	00.00 ± 00.00	0.42	1.82 ± 1.41	1.05 ± 0.82	1.24 ± 0.78	0.003
LDL-c	3.87 ± 0.97	3.48 ± 0.94	3.24 ± 1.00	0.29	3.79 ± 0.97	4.09 ± 0.94	00.00 ± 00.00	0.51	3.77 ± 0.74	3.89 ± 1.09	2.72 ± 1.25	0.11

Predictors (constants): *TLR4* gene (rs11536889, rs4986790, and rs4986791). Dependent variables are listed in covariates.

**Table 7 cells-12-02340-t007:** Haplotype association between the genetic variants present in *TLR4* gene in combination of both T2DM and control subjects.

S. No	rs11536889	rs4986790	rs4986791	Freq	OR (95%CI)	*p*-Value
1	G	A	C	0.7367	1.00	1.00
2	G	A	T	0.199	4.27 (2.02–9.01)	<0.0001
3	A	A	C	0.0325	8.46 (1.40–51.12)	0.021
4	A	A	T	0.0127	6.42 (0.58–71.52)	0.13

Freq—frequency; OR—odds ratio; 95%CI—95% confidence interval.

**Table 8 cells-12-02340-t008:** Performance of linkage disequilibrium analysis in combined T2DM cases and controls.

Subjects	L1	L2	D’	r^2^
All cases	rs11536889	rs4986790	0.217	0.014
All cases	rs11536889	rs4986791	0.029	0.02
All cases	rs4986790	rs4986791	0.126	0.001

L1—line 1; L2—line 2; D’—disequilibrium value.

**Table 9 cells-12-02340-t009:** Gene–gene interaction to determine the risk of T2DM patients.

Model No	Best Combination of Genes	Training Accuracy	Testing Accuracy	CVC	*p*-Value	Total Sensitivity	Total Specificity	X^2^	OR (95%CI)	F-Measure	Kappa
1	rs11536889	0.6476	0.6476	10/10	<0.0001	0.5619	0.6476	18.86	3.52 (1.97–6.29)	0.6146	0.2952
2	rs11536889, rs4986791	0.6857	0.6857	10/10	<0.0001	0.6571	0.7143	26.16	4.79 (2.67–8.59)	0.6765	0.3714
3	rs11536889, rs4986790, rs4986791	0.6905	0.6905	10/10	<0.0001	0.6762	0.7048	30.51	4.98 (2.77–8.93)	0.686	0.381

CVC = cross-validation consistency; X^2^ = chi-square value.

## Data Availability

The data can be obtained from the corresponding author upon reasonable request.

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
