# Peer review of "Molecular Effect of Variants in Toll-like Receptor 4 Gene in Saudi Patients with Type 2 Diabetes Mellitus"

_cells, 2023, doi:10.3390/cells12192340_

Round 1

Reviewer 1 Report

The authors mainly confirmed the results of previous report that the two SNPs are associated with T2DM in the Saudi population and the relationship between HDLc levels and the TLR4 gene, but this present work is lack of novelty for a publication in the Journal, Cells.

Major points

1. The authors should do more investigations on the mechanisms whereby SNPs affect the genes' expression associated with type 2 diabetes.

2. The Figure 1, Table1,Figure 2  and Table 2 should be presented as supplemental files since these are only the raw data.

3. The authors should simplify the discussion and just focus on the main findings in the Discussion section.

Minor points

There are many unambiguous or wrong presentations. Such as,

1. In line 76, whats the meaning of single amino acid substitutions’ in the sentence ‘this study was designed to analyze single amino acid substitutions SNPs....?

2. The title of Table 6 should be changed. And the all the Table legends should be added more details.

3. The title of Fig.3 should be rewritten and more details added to the figure legend.

Author Response

The authors mainly confirmed the results of previous report that the two SNPs are associated with T2DM in the Saudi population and the relationship between HDLc levels and the TLR4 gene, but this present work is lack of novelty for a publication in the Journal, Cells.

Dear Reviewer,

Thank you for your valuable comments. We have screened 3 major variants present in TLR4 gene in T2DM in the Saudi population. Diabetes is becoming more common, and genetic variables are leading to consanguineous marriages, which we attempted to explore in our group.

Major points

  1. The authors should do more investigations on the mechanisms whereby SNPs affect the genes' expression associated with type 2 diabetes.
  2. A) We have updated in the revised manuscript. The role of TLR4 gene is mainly connected with pancreas and adipose tissue and also it will affect the immune response.
  3. The Figure 1, Table1,Figure 2 and Table 2 should be presented as supplemental files since these are only the raw data.
  4. A) Dear Reviewer, we agree with Table-1, Figures 1-2 but we request to continue Table-2 in the main document as it represents the demographic details present in the manuscript.
  5. The authors should simplify the discussion and just focus on the main findings in the Discussion section.
  6. A) All the authors have reworked on the discussion and removed some of the unwanted sentences.

Minor points

There are many unambiguous or wrong presentations. Such as,

  1. In line 76, what’s the meaning of ‘single amino acid substitutions’ in the sentence ‘this study was designed to analyze single amino acid substitutions SNPs....’?
  2. A) To avoid the plagiarism, we have rewritten the above sentence. The single amino acid substitution indicates there is only one modification in the protein sequence.
  3. The title of Table 6 should be changed. And the all the Table legends should be added more details.
  4. A) Title of the Table-6 was updated. Table legends were also added in detailed.
  5. The title of Fig.3 should be rewritten and more details added to the figure legend.
  6. A) Title was updated and figure legend was added.

Reviewer 2 Report

Type 2 diabetes is a pandemic disease characterized by hyperglycemia, ineffective insulin use, and insulin resistance. Inflammation-related insulin resistance is thought to play an important role in the etiology of the disease. TLR4 is the central receptor of the natural immune system and has an important role as a trigger of the inflammatory response. 

Because toll-like receptor 4 (TLR4) plays  important roles in cellular immunity and TLR4 polymorphisms have been shown to be associated with suscetibility to a range of diseases, the present study aimed to investigate the association between TLR4 gene polymorphisms and the incidence of type 2 diabetes mellitus (T2DM).

The presented study includes multiple aspects related to: demographic profile for performing student t-tests between T2DM and control subjects, allelic association studies, analysis of ANOVA examined between dependent and independent variables, haplotype interference analysis, association of gene-gene interaction analysis studied via GMDR model.

The study presented significant differences in the allelic distributions of rs11536889 and rs4986791 SNPs between patients with T2DM and controls.

In this study, 14 covariates were identified as dependent variables, including age, weight, BMI, waist circumference, hip circumference, SBP, DBP, FBG, HbA1c, TG, TC, HDLc, LDLc. 

The study there was no statistically significant association between the dependent or independent variables in the rs11536889 SNP.  The waist circumference was strongly associated with rs4986790 SNP. The hip circumference, SBP, DBP and HDLc levels were associated with the rs4986791 SNP. 

The results confirmed that rs11536889 and rs4986790 SNPs are genotypically and allelically associated with T2DM in Saudi patients, and additional statistical analysis confirmed the relationship between HDLc levels and the TLR4 gene. 

The article is well written and documented. The bibliography presented includes recent titles. I recommend it for publication. 

Author Response

Type 2 diabetes is a pandemic disease characterized by hyperglycemia, ineffective insulin use, and insulin resistance. Inflammation-related insulin resistance is thought to play an important role in the etiology of the disease. TLR4 is the central receptor of the natural immune system and has an important role as a trigger of the inflammatory response. Because toll-like receptor 4 (TLR4) plays important roles in cellular immunity and TLR4 polymorphisms have been shown to be associated with susceptibility to a range of diseases, the present study aimed to investigate the association between TLR4 gene polymorphisms and the incidence of type 2 diabetes mellitus (T2DM).The presented study includes multiple aspects related to: demographic profile for performing student t-tests between T2DM and control subjects, allelic association studies, analysis of ANOVA examined between dependent and independent variables, haplotype interference analysis, association of gene-gene interaction analysis studied via GMDR model. The study presented significant differences in the allelic distributions of rs11536889 and rs4986791 SNPs between patients with T2DM and controls. In this study, 14 covariates were identified as dependent variables, including age, weight, BMI, waist circumference, hip circumference, SBP, DBP, FBG, HbA1c, TG, TC, HDLc, LDLc. The study there was no statistically significant association between the dependent or independent variables in the rs11536889 SNP.  The waist circumference was strongly associated with rs4986790 SNP. The hip circumference, SBP, DBP and HDLc levels were associated with the rs4986791 SNP.  The results confirmed that rs11536889 and rs4986790 SNPs are genotypically and allelically associated with T2DM in Saudi patients, and additional statistical analysis confirmed the relationship between HDLc levels and the TLR4 gene. The article is well written and documented. The bibliography presented includes recent titles. I recommend it for publication. 

A) Dear Reviewer, Thank you for your valuable comments. We really appreciate it and thank you for recognizing our hard work towards this manuscript.

Reviewer 3 Report

The manuscript entitled “Relevance of genetic variants in Toll like receptor 4 gene in Saudi patients with type 2 diabetes mellitus" appears to be interesting, but there are many flaws and concerns on it. Study can be greatly improved if following suggestions were incorporated.

1.      The title of the paper is not accurately expressed, and I think it needs to be rewritten.

2.      Some references missing. For example, “Diabetes Mellitus (DM) is a clinical syndrome characterized by hyperglycemia at 32 different states, such as fasting, post-prandial, and oral glucose tolerance tests. The impact 33 of blood glucose levels on human health is often underestimated.”, and etc.

The following reference may increase the reader’s comprehension:

Sheykhhasan M. Towards Standardized Stem Cell Therapy in Type 2 Diabetes Mellitus: A Systematic Review. Curr Stem Cell Res Ther. 2019;14(1):75-76. doi: 10.2174/1574888X1401181217125608. PMID: 30684321.

3.      In order to make the paper more interesting to read, I suggested that the authors could add one graphical abstract to the manuscript.

Paper is replete with grammatical mistakes. Needs rewriting and thorough evaluation.

Author Response

The manuscript entitled “Relevance of genetic variants in Toll like receptor 4 gene in Saudi patients with type 2 diabetes mellitus" appears to be interesting, but there are many flaws and concerns on it. Study can be greatly improved if following suggestions were incorporated.

Dear Reviewer,

Thank you for your comments. We have justified the raised comments below. This manuscript was edited with native speakers and then submitted. Please find the attachment of the edited certificate from EDITAGE.

  1. The title of the paper is not accurately expressed, and I think it needs to be rewritten.
  2. A) Title has been updated in the revised manuscript.
  3. Some references missing. For example, “Diabetes Mellitus (DM) is a clinical syndrome characterized by hyperglycemia at 32 different states, such as fasting, post-prandial, and oral glucose tolerance tests. The impact 33 of blood glucose levels on human health is often underestimated.”, and etc. The following reference may increase the reader’s comprehension:

Sheykhhasan M. Towards Standardized Stem Cell Therapy in Type 2 Diabetes Mellitus: A Systematic Review. Curr Stem Cell Res Ther. 2019;14(1):75-76. doi: 10.2174/1574888X1401181217125608. PMID: 30684321.

  1. A) Thank you for your valuable suggestion, now, we have added the recommended reference in the revision.
  2. In order to make the paper more interesting to read, I suggested that the authors could add one graphical abstract to the manuscript.
  3. A) Graphical abstract is added in the revised manuscript.

Paper is replete with grammatical mistakes. Needs rewriting and thorough evaluation.

Manuscript was already edited with English editing services. However, careful revision was carried out to rectify the errors.

Round 2

Reviewer 1 Report

The manuscript is acceptable since the authors have verified almost all of the questions.

The manuscript has been improved much in the English writing.

Reviewer 3 Report

The authors completely responded to the observations made previously. They also inserted useful and clarifying information. Thank you for editing the manuscript. Now, the manuscript is much better than the former.